

# Identification of CDT1 as a prognostic marker in human lung adenocarcinoma using bioinformatics approaches

Jing Jiang[1,2,*], Yu Zhang[3,*], Jun Wang[4], Xuefei Yang[1], Xingchang Ren[3], Hai Huang[4], Jue Wang[1], Jinhua Lu[1], Yazhen Zhong[1], Zechen Lin[1], Xianlei Lin[1], Yewei Jia[5] and Shengyou Lin[6]

[1] Department of Oncology, Hangzhou Traditional Chinese Medicine (TCM) Hospital Affiliated to Zhejiang Chinese Medical University, Hangzhou, China
[2] The Third Clinical Medical College, Zhejiang Chinese Medical University, Hangzhou, China
[3] Department of Pathology, Hangzhou Traditional Chinese Medicine (TCM) Hospital Affiliated to Zhejiang Chinese Medical University, Hangzhou, China
[4] Department of Cardiothoracic Surgery, Hangzhou Traditional Chinese Medicine (TCM) Hospital Affiliated to Zhejiang Chinese Medical University, Hangzhou, China
[5] Department of Internal Medicine 3, Friedrich-Alexander-University Erlangen-Nürnberg (FAU) and Universitätsklinikum Erlangen, Erlangen, Germany
[6] Department of Oncology, The First Affiliated Hospital of Zhejiang Chinese Medical University (Zhejiang Provincial Hospital of Chinese Medicine), Hangzhou, China
* These authors contributed equally to this work.

Corresponding authors
Yewei Jia, jiayewei93@gmail.com
Shengyou Lin,
shengyoulin@zcmu.edu.cn

## ABSTRACT

**Background:** Lung cancer has the highest cancer-related mortality worldwide. Lung adenocarcinoma (LUAD) is the most common histological subtype of non-small cell lung cancer (NSCLC). Chromatin licensing and DNA replication factor 1 (CDT1), a key regulator of cell cycle control and replication in eukaryotic cells, has been implicated in various cancer-related processes. Given its significant role in cancer, the focus on CDT1 in this study is justified as it holds promise as a potential biomarker or therapeutic target for cancer treatment. However, its prognostic value in lung adenocarcinoma (LUAD) remains unclear.

**Methods:** Bioinformatics analysis was conducted using data obtained from The Cancer Genome Atlas (TCGA) and Gene Expression Omnibus (GEO) databases. Gene Ontology (GO) and Kyoto Encyclopedia of Genes and Genomes (KEGG) databases were utilized to predict biological processes and signaling pathways, respectively. The LinkedOmics database was employed to identify differentially expressed genes (DEGs) associated with CDT1. Nomograms and Kaplan-Meier plots were generated to assess the survival rates of patients with lung adenocarcinoma (LUAD). To determine the RNA and protein expression levels of CDT1 in LUAD and adjacent normal tissues, quantitative polymerase chain reaction (qPCR) and immunohistochemistry techniques were employed, respectively.

**Results:** CDT1 was upregulated in the vast majority of cancer tissues, based on pan-cancer analysis in TCGA and GEO datasets, as to lung cancer, the level of CDT1 expression was much higher in LUAD tissue than in healthy lung tissue. Our clinical data supported these findings. In our study, we used a specific cutoff value to dichotomize the patient samples into high and low CDT1 expression groups. The Kaplan–Meier survival curve revealed poor survival rates in CDT1 high expression group than the low expression group. To determine if CDT1 expression

was an independent risk factor in LUAD patients, univariate and multivariate Cox regression analyses were performed. The result showed that CDT1 was a potential novel prognosis factor for LUAD patients, whose prognosis was poorer when CDT1 expression was higher. Based on functional enrichment analysis, highly expressed DEGs of CDT1-high patients were predicted to be involved in the cell cycle. According to our analysis of immune infiltration, CDT1 exhibited a strong correlation with specific immune cell subsets and was found to be a significant predictor of poor survival in patients with LUAD.

**Conclusions:** Our research found that CDT1 was upregulated in LUAD and that high CDT1 expression predicted poor prognosis. We comprehensively and systematically analyzed the expression level in the datasets as well as in our own clinical samples, we also evaluated the prognostic and diagnostic value of CDT1, and finally, the potential mechanisms of CDT1 in the progression of LUAD. These results suggested that CDT1 may be a prognostic marker and therapeutic target for LUAD.

## INTRODUCTION

Non-small cell lung cancer (NSCLC), which accounts for 85% of lung cancers and the majority of lung cancer deaths, is the most prevalent type of lung cancer and continues to be the most common human malignancy worldwide (*Kim & Giaccone, 2018*). Lung adenocarcinoma (LUAD), lung squamous cell carcinoma, and large cell lung carcinoma are three histologically distinct kinds of NSCLC. LUAD accounts for approximately 40% of all NSCLC cases among these three cancer types (*Testa, Castelli & Pelosi, 2018*). Despite improvements in LUAD treatment strategies and the introduction of numerous therapeutic methods in recent decades, the 5-year survival rate of patients with LUAD remains very poor (*Siegel, Miller & Jemal, 2019*). Therefore, an urgent need to develop novel therapeutic strategies and prognostic markers for LUAD is apparent.

A gene called chromatin licensing and DNA replication factor 1 (CDT1) has previously been identified as being essential for the initiation of DNA replication, as well as for the replication and control of eukaryotic cells (*Yang et al., 2019*). CDT1 is also thought to be strongly associated with tumorigenesis. CDT1 overexpression has been linked to lower survival and prognosis rates in several tumor forms (*Mahadevappa et al., 2017*; *Seo et al., 2005*). In addition, CDT1 overexpression alone can induce overt re-replication in some cancer-derived cells (*Vaziri et al., 2003*; *Sugimoto et al., 2008*, *2009*). However, the precise roles and prognostic relevance of CDT1 in the evolution of LUAD are still unknown. Knowing CDT1 function is essential for us to better understand cancer transformation and progression.

In the present study, we comprehensively explored CDT1's expression patterns, potential function, prognostic value, and relationship with immune infiltration in LUAD through bioinformatics analyses. Clinical tissues were also used to verify expression levels of CDT1 in LUAD.

# MATERIALS AND METHODS

## Data acquisition

High-throughput RNA sequencing data and clinical information regarding lung cancer patients were obtained using the UCSC Xena browser (version: 2019-07-20, http://xenabrowser.net/datapages/). For external validation, transcriptome profiling data from patients with LUAD from the Gene Expression Omnibus datasets GSE118370, GSE140797 and GSE31210 were used.

R software was used to examine the levels of CDT1 expression in LUAD and other malignancies (version 4.0.4). R was used to perform data analyses in this study (version 4.0.4). All the raw codes used in the manuscript could be found at https://github.com/YeweiJia/CDT1-Jiang.

## LinkedOmics analysis

The LinkedOmics database (http://www.linkedomics.org/login.php) contains multi-omics data and clinical information from the TCGA portal for 32 cancer types (*Vasaikar et al., 2018*). LinkedOmics is an online platform that integrates multi-omics data from TCGA to enable comprehensive analysis. We used the linkFinder module to evaluate DEGs related to CDT1.

## Kaplan-Meier survival analysis

The Kaplan-Meier method is a widely used statistical technique for estimating the survival probability or survival function of a given population over time. In this study, we employed the Kaplan-Meier method to analyze the survival data of lung adenocarcinoma (LUAD) patients. The Kaplan-Meier estimator calculates the survival probability at each time point and generates a survival curve. The survival curve displays the proportion of patients surviving at different time intervals. The log-rank test or other appropriate statistical tests were performed to compare the survival curves between different groups. The $p$-value obtained from the statistical test provides information about the significance of the differences in survival between groups.

## Identification of differentially expressed genes and building a nomogram

The R "Limma" package was applied to standardize the read counts and perform differential gene analysis with the threshold of LogFC >1.5 and <−1.5. The R'rms'package and 'survival'package were used to establish a prognostic nomogram to predict the survival risk.

Nomogram is a statistical tool that integrates multiple prognostic factors to estimate the probability of a specific outcome. We used a nomogram in our study to interpret and utilize the predictive model for assessing risk or prognosis based on specific patient characteristics or biomarkers.

## Human tissue specimens

To investigate CDT1 levels in human LUAD, we obtained tumor tissues and paired adjacent non-tumorous tissues during radical resection of patients without prior chemotherapy or radiotherapy at the Department of Cardiothoracic Surgery, Hangzhou Traditional Chinese Medicine Hospital, affiliated with Zhejiang Chinese Medical University, from January 2022 to July 2022. Resected LUAD-adjacent non-tumor samples and matched tumor tissues were obtained and instantly stored in liquid nitrogen (Table 1). This study was approved by the ethics committee of the Hangzhou Traditional Chinese Medicine Hospital, affiliated with Zhejiang Chinese Medical University (Hangzhou, China; No. 2021LH005), and written consent was obtained from all patients. The grades and histological types of all tissue samples were independently verified by two professional pathologists.

## RNA extraction and qPCR

Total RNA was extracted from the obtained tissues using TRIrizol reagent (Takara Bio, Shiga, Japan). cDNA was synthesized using the PrimeScript™ RT Master Mix (Takara Bio, Shiga, Japan). SYBR Premix Ex Taq™ II with a PCR detection system (Takara Bio, Shiga, Japan) was used to investigate the expression of the indicated genes. Transcriptional levels were normalized to those of the internal control gene, β-actin. The following primers were used: CDT1 forward, 5′-GGAGGTCAGATTACCAGCTCAC-3′ and reverse, 5′-TTGAC GTGCTCCACCAGCTTCT-3′; β-actin forward, 5′-CATCCGCAAAGACCTGTACG-3′ and reverse, 5′-CCTGCTTGCTGATCCACATC-3′. The PCR was run as follows: initial denaturation at 95 °C for 35 s as an initial denaturation step, amplification for 40 cycles with denaturation at 95 °C for 5 s, and annealing at 60 °C for 30 s. The melting curve analysis was performed at the end of the PCR cycle. All reaction information was collected using the Applied Biosystems 7500 Real-Time PCR System (Thermo Scientific, Waltham, MA, USA), and normalized expression was calculated as the relative fold-change, using the formula $2^{-\Delta\Delta CT}$.

## Hematoxylin and eosin staining

The LUAD tissues and paired adjacent normal tissues were preserved in 4% paraformaldehyde, embedded in paraffin blocks, sliced at a thickness of 4 μm, dewaxed, dehydrated sequentially, and then stained with hematoxylin and eosin (H&E). Ultimately, all slides were independently examined and pathologically analyzed by two pathologists (Department of Pathology, Hangzhou Traditional Chinese Medicine Hospital, affiliated with Zhejiang Chinese Medical University), under an optical microscope (Olympus, Tokyo, Japan).

## Immunohistochemistry and scoring analyses

Immunohistochemistry was conducted to determine CDT1 protein expression and prognostic value. LUAD tissues and paired adjacent normal tissues were stained by immunohistochemistry with anti-human CDT1 (1:300; Rabbit; 14382-1-AP; Proteintech, Rosemont, IL, USA), followed by IVision™ Poly-HRP goat anti-mouse/rabbit secondary

**Table 1 The sixteen patients' characteristics of the fresh frozen samples.**

| | Age | Gender | Smoking history | Location | Surgery | TNM | Histology/ predominant growth patterns | TTF-1 | NapsinA | CK7 | Bronchial involvement/ lymphovascular invasion | Status |
|---|---|---|---|---|---|---|---|---|---|---|---|---|
| 1 | 76 | M | No | LLL | VATS radical resection of left lower lung cancer | pT2aN1M0 IIB | Adenocarcinoma/ solid+acinar +papillary | (+) | (+) | (+) | No | Alive |
| 2 | 73 | M | No | RUL | VATS radical resection of right upper lung cancer | pT3N0M0 IIB | Adenocarcinoma/ acinar | (+) | (+) | (+) | No | Alive |
| 3 | 67 | F | No | LLL | VATS radical resection of left lower lung cancer | pT1cN0M0 IA3 | Adenocarcinoma/ papillary +acinar +adherence | (+) | (+) | (+) | No | Alive |
| 4 | 51 | M | No | RML | VATS partial resection of middle lobe of right lung | pT1aN0M0 IA1 | Adenocarcinoma/ acinar +adherence +papillary | (+) | (+) | (+) | No | Alive |
| 5 | 34 | F | No | LUL | VATS partial resection of lingual segment of left upper lobe of lung | pTisN0M0 IA | Adenocarcinoma/ adherence | NA | NA | NA | No | Alive |
| 6 | 58 | M | No | LUL | VATS Left upper apical posterior segment resection | pT1aN0M0 IA1 | Adenocarcinoma/ adherence | (+) | (+) | (+) | No | Alive |
| 7 | 70 | M | No | LUL | VATS radical resection radical resection of left upper lung cancer | pT1bN0M0 IA2 | Adenocarcinoma/ acinar | (+) | (+) | (+) | No | Alive |
| 8 | 48 | M | No | RUL | VATS partial resection of right upper lobe of lung | pT1aN0M0 IA1 | Adenocarcinoma/ adherence | (+) | (+) | (+) | No | Alive |
| 9 | 46 | M | No | RUL | VATS resection of anterior segment of right upper lobe of lung | pT1aN0M0 IA1 | Adenocarcinoma/ adherence | (+) | (+) | (−) | No | Alive |
| 10 | 74 | M | No | RLL | VATS partial resection of right lower lobe of lung | pT1bN0M0 IA2 | Adenocarcinoma/ adherence +acinar +papillary | (+) | (+) | (−) | No | Alive |
| 11 | 58 | F | No | RLL | VATS radical resection of right lung | pT1bN0M0 IA2 | Adenocarcinoma/ acinar +papillary +adherence | (+) | (+) | (+) | No | Alive |
| 12 | 70 | M | Yes | RLL | VATS radical resection of right upper lung cancer | pT1cN0M0 IA3 | Adenocarcinoma/ acinar+solid | (+) | (+) | (+) | No | Alive |
| 13 | 76 | F | No | RUL | VATS partial resection of right upper lobe of lung | pT1bN0M0 IA2 | Adenocarcinoma/ adherence | (+) | (+) | (+) | No | Alive |

(Continued)

| | Age | Gender | Smoking history | Location | Surgery | TNM | Histology/predominant growth patterns | TTF-1 | NapsinA | CK7 | Bronchial involvement/lymphovascular invasion | Status |
|---|---|---|---|---|---|---|---|---|---|---|---|---|
| 14 | 74 | F | No | RUL | VATS radical resection of right upper lung cancer | pT1bN0M0 IA2 | Adenocarcinoma/acinar | (+) | (+) | (+) | No | Alive |
| 15 | 47 | M | No | LUL | VATS radical resection of left upper lung cancer | pT1bN0M0 IA2 | Adenocarcinoma/adherence | (+) | (+) | (+) | No | Alive |
| 16 | 61 | M | Yes | RUL | VATS radical resection of right upper lung cancer | pT1cN0M0 IA3 | Adenocarcinoma/acinar | (+) | (+) | (+) | No | Alive |

**Note:**
F, female; M, male; LLL, left lower lobe; LUL, left upper lobe; RUL, right upper lobe; RLL, right lower lobe; RML, right middle lobe; VATS, video-assisted thoracoscopic surgery; TNM, Tumor location, tumor size, differentiation status, invasion depth, lymph node metastasis, distant metastasis and tumor-node metastasis stage; TTF-1, positive thyroid transcription factor-1; CK7, Cytokeratin 7; and NA, no application.

antibody reagent (DD23-100; Xiamen Talent Biomedical Technology Co., Ltd., Xiamen, China) and DAB treatment. Then, the samples were observed under a microscope and photographed for further analysis. Each LUAD sample was evaluated based on staining intensity and positively stained cell percentage. The H-score was calculated as: (percentage of weak-intensity cells × 1) + (percentage of moderate-intensity cells × 2) + (percentage of strong-intensity cells × 3). Numbers 0, 1, 2, and 3 indicated positive cells. H-score values ranged between 0 and 3. A paired t-test was used to compare CDT1 expression between LUAD tissues and the paired non-tumor tissues.

## GO and KEGG enrichment analyses

Gene ontology (GO) (http://www.geneontology.org/) and Kyoto Encyclopedia of Genes and Genomes (KEGG) (http://www.genome.jp/kegg/) pathways that are usually used to predict the biological processes as well as signaling pathways, in which some genes could participate, analyses were implemented using the hypergeometric test method, and an adjusted $P$-value $< 0.05$ was considered significant by Fisher's exact test.

## Tumor immune infiltration analysis

The R 'ESTIMATE', 'CIBERSORT' (Chen et al., 2018), and 'GSVA' (Hänzelmann, Castelo & Guinney, 2013) packages were used to analyze the immune infiltration of TME.

## Statistical analysis

All statistical analyses were performed in R (v4.0.4) and Prism 8.0 (GraphPad Software Inc., San Diego, CA, USA). Kaplan–Meier curve and log-rank test were used to evaluate differences in survival rate between the groups. The student's t-test or Wilcoxon test was performed for continuous data, and the Chi-square test or Fisher's exact test was performed for categorical variables. Univariate and multivariate analyses using Cox
proportional hazard modeling were performed to estimate the risk of death. Effects were considered significant if $P < 0.05$.

# RESULTS

## CDT1 expression was upregulated in LUAD, based on the TCGA and GEO databases

CDT1 was found to be highly elevated in a variety of malignancies when compared to normal tissues, according to a pan-cancer analysis utilizing Wilcoxon rank-sum and Wilcoxon signed-rank tests (Fig. 1A, $P < 0.001$). In order to show the expression of CDT1 in LUAD more clearly, the TCGA database's paired tumor and normal nearby samples, as well as the unpaired samples, were also studied using the two different statistical analysis techniques. Results showed that the tumor group had a higher CDT1 expression than the normal group (Fig. 1B). Paired tumor tissues and normal tissues from two GEO datasets (GSE118370 and GSE140797) verified that CDT1 was upregulated in tumor tissues *versus* normal tissues (Fig. 1C), in addition, another cohort (GSE31210) indicated high expression level of CDT1 in LUAD sample compared to normal samples (Fig. 1C). Then, volcano plots of the three GEO datasets indicated the expression level of CDT1 among differentially expressed genes (DEGs) of the datasets (Fig. 1D).

Together, these results indicated CDT1 was significantly up-regulated in LUAD.

## Correlation between CDT1 expression and poor prognosis

Based on the TCGA dataset from the UCSC Xena browser (version: 2019-07-20, http://xenabrowser.net/datapages/), the relationship between CDT1 expression and the prognosis of LUAD patients was assessed. The survival rate of patients with LUAD was calculated using R program software. The results showed that no difference in the survival rate between the male and female patients was observed (Fig. 2A). However, based on the cutoff (Fig. 2B), the high CDT1 expression group had considerably higher accumulated risk and worse OS than the low CDT1 expression group (log-rank $P < 0.001$; Figs. 2C and 2D). In addition, we also used GEO cohort to verify the survival rate, the data demonstrated that CDT1 high-expression group had a poorer survival rate compared to low-expression group (Fig. 2E).

In addition, in order to incorporate these prognostic factors, we constructed a nomogram that was used to predict or assess the risk or prognosis (*Hartaigh et al., 2018*) based on all the independent prognostic indicators for OS (Fig. 2F). Nomogram is a graphical tool used for predicting a specific outcome or event based on multiple variables. The nomogram visually represents a mathematical model that combines the effects of different predictor variables to estimate a particular outcome. Each predictor variable is assigned a numerical value, and by drawing a vertical line from each variable's value to the points axis, the corresponding points for each variable are determined. Based on our results, we provided several predictors, including disease stages, age, and different expression levels of CDT1, and so on to establish a nomogram, a high total score predicted a high survival risk, whereas a low total score showed the opposite. Our data indicated in either in TCGA cohort or GSE31210, CDT1 high-expression group was deeply associated

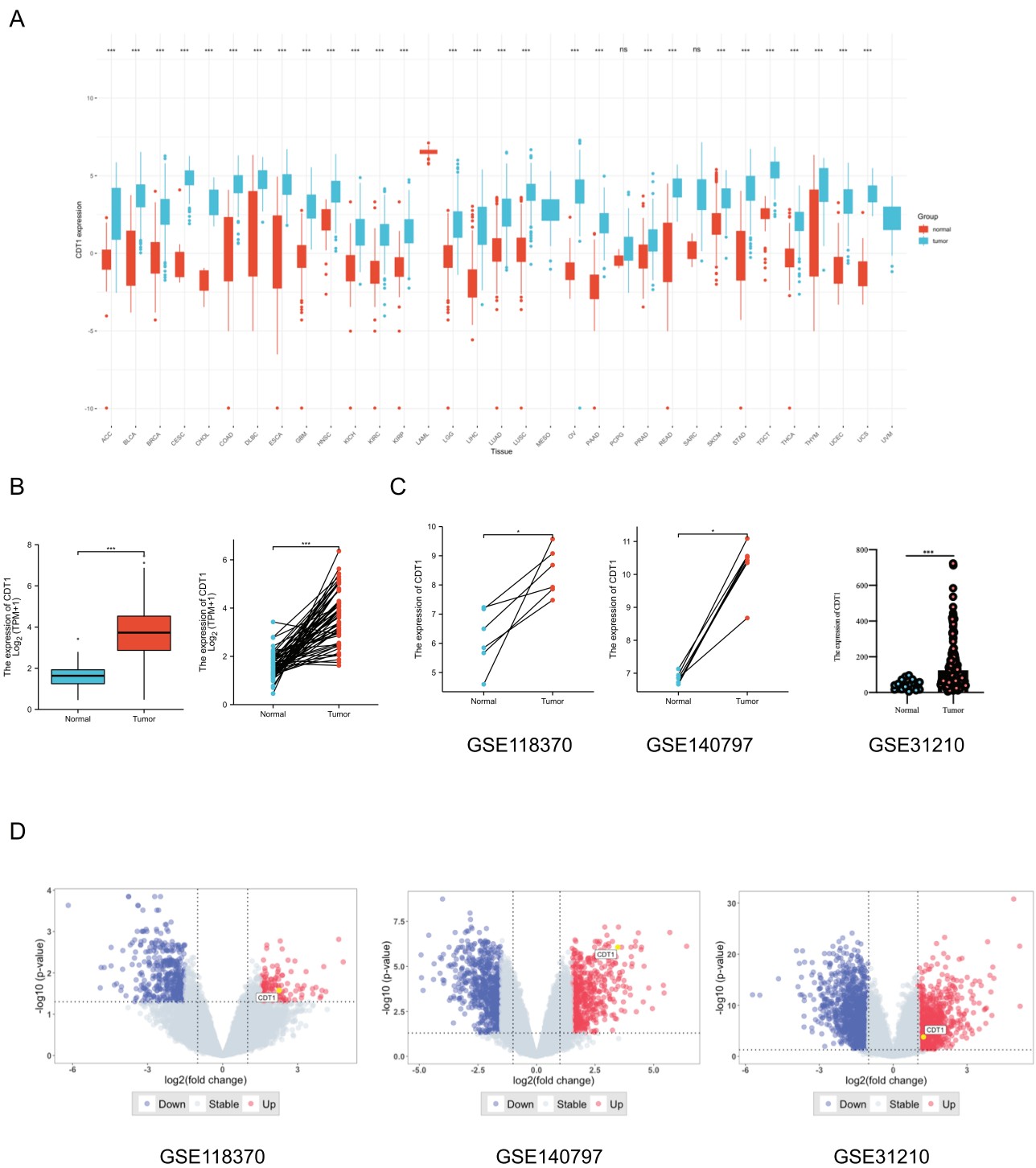

**Figure 1  CDT1 expression is upregulated in LUAD, based on the TCGA and GEO databases.** (A) The expression level of CDT1 in pan-cancer (Blue: tumor tissue; Red: normal tissue). (B) The expression of CDT1 in LUAD and normal tissues (left: non-paired, $n = 30$; right: paired, $n = 30$) from the TCGA dataset (Blue: normal tissue; Red: tumor tissue). (C) The expression level of CDT1 in LUAD and normal tissues (paired and non-paired) from the GEO datasets (Blue: normal tissue; Red: tumor tissue. GSE118370, $n = 6$; GSE140797, $n = 5$; GSE31210, normal sample, $n = 20$, tumor sample, $n = 226$). (D) Volcano plots of GEO datasets (GSE118370, GSE140797 and GSE31210) (Blue: down-regulated genes; Red: up-regulated genes) (*$P < 0.05$, ***$P < 0.001$). CDT1, chromatin licensing and DNA replication factor 1; TCGA, The Cancer Genome Atlas; GEO, Gene Expression Omnibus; LUAD, lung adenocarcinoma.

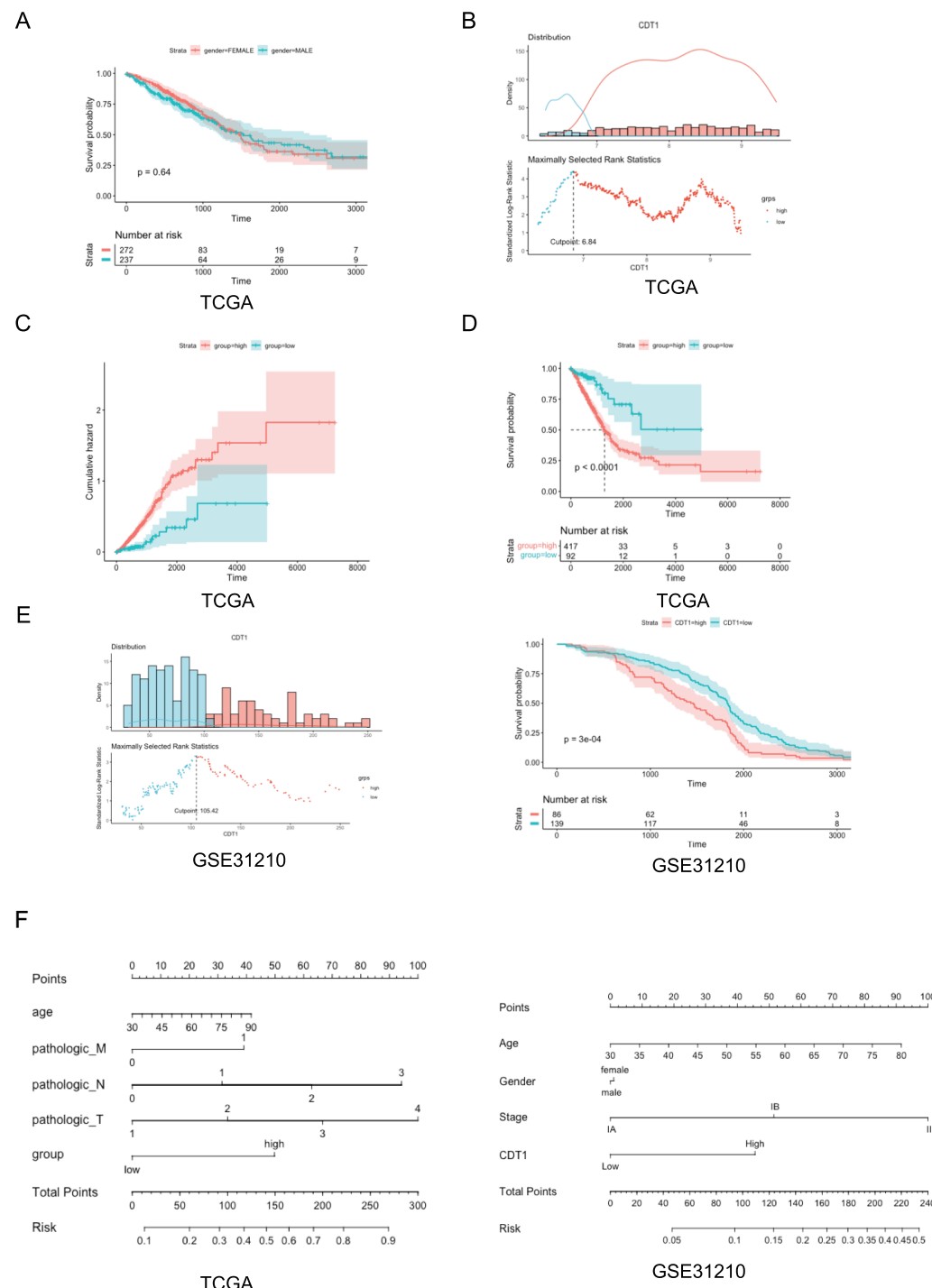

**Figure 2 Correlation between CDT1 expression and poor prognosis.** (A) The Kaplan-Meier curve of OS in patients of different genders in TCGA cohort (Blue: male, *n* = 237; Red: female, *n* = 272). (B) The distribution of CDT1 expression in LUAD (Blue: CDT1 low, *n* = 92; Red: CDT1 high, *n* = 417). (C) Cumulative hazard for CDT1 high-expression and low-expression group (Blue: CDT1 low, *n* = 92; Red: CDT1 high, *n* = 417). (D) KM curves for overall survival (Blue: CDT1 low, *n* = 92; Red: CDT1 high, *n* = 417). (E) The distribution of CDT1 expression in GSE31210 (Blue: CDT1 low, *n* = 139; Red: CDT1 high, *n* = 86), and KM curves for overall survival (Blue: CDT1 low, *n* = 139; Red: CDT1 high, *n* = 86). (F) Nomogram based on CDT1 expression and clinicopathological factors (Including TCGA and GSE31210,

**Figure 2** (continued)

for TCGA, *n* = 509, for GSE31210, *n* = 225). CDT1, chromatin licensing and DNA replication factor 1; OS, overall survival; TCGA, The Cancer Genome Atlas. For the survival: The shadows in Figure 2 represent the confidence intervals around the survival estimates. The shadows are typically shown as shaded areas around the plotted survival curves and provide information about the uncertainty associated with the survival estimates. "Number at risk" tables: The "Number at risk" tables accompanying the survival curves in Figure 2 represent the number of individuals or cases still under observation at each time point. For the Cumulative hazard: The cumulative hazard represents the accumulation of the hazard rates (instantaneous event rates) up to a certain point in time, the x-axis represents time, while the y-axis represents the cumulative hazard values. Each curve in the figure corresponds to a different subgroup or condition, allowing a visual comparison of the cumulative risk profiles. For nomogram: Nomogram is a graphical tool used for predicting a specific outcome or event based on multiple variables. The nomogram visually represents a mathematical model that combines the effects of different predictor variables to estimate a particular outcome. Each predictor variable is assigned a numerical value, and by drawing a vertical line from each variable's value to the points axis, the corresponding points for each variable are determined.               

with poor survival rate. These results indicate that LUAD with increased CDT1 expression is associated with poor OS.

Taken together, these data revealed high expression level of CDT1 was positively associated with a poor survival rate.

## Clinical samples to verify the expression level of CDT1 in LUAD and paired adjacent normal samples

Quantitative real-time PCR (qPCR) was first performed to evaluate the expression levels of CDT1 in LUAD and paired adjacent normal samples. Sixteen patient samples were used in the experiment. Patient information is shown in Table 1. The data showed that CDT1 was more highly expressed in the tumor group than in the normal group (Fig. 3A). H&E staining results are shown in Fig. 3B. Immunohistochemistry was then performed to explore the expression levels of CDT1 in LUAD and paired normal samples. Immunohistochemistry results indicated that the cytoplasmic staining level of CDT1 in tumor tissues is higher than that in adjacent normal tissues (Fig. 3C).

Our clinical data, including the qPCR and IHC, verified the high expression level of CDT1 in LUAD compared to paired adjacent normal tissue.

## CDT1-related functional enrichment analysis

We first divided CDT1 into two groups according to the distribution, the R 'limma' package was used to analyze the Differentially Expressed Genes (DEGs). A total number of 466 DEGs were identified, including 417 upregulated genes and 49 down-regulated genes (Fig. 4A). GO and KEGG pathway enrichment analyses were then performed to analyze the functional enrichment. The GO analyses (Fig. 4B) indicated that the upregulated DEGs were mainly enriched in organelle fission, DNA replication, extracellular matrix, ATPase activity, and so on. The KEGG analysis (Fig. 4B) indicated the DEGs were primarily enriched in the Cell cycle, Carbon metabolism, DBA replication, p53 signaling pathway, and so on.

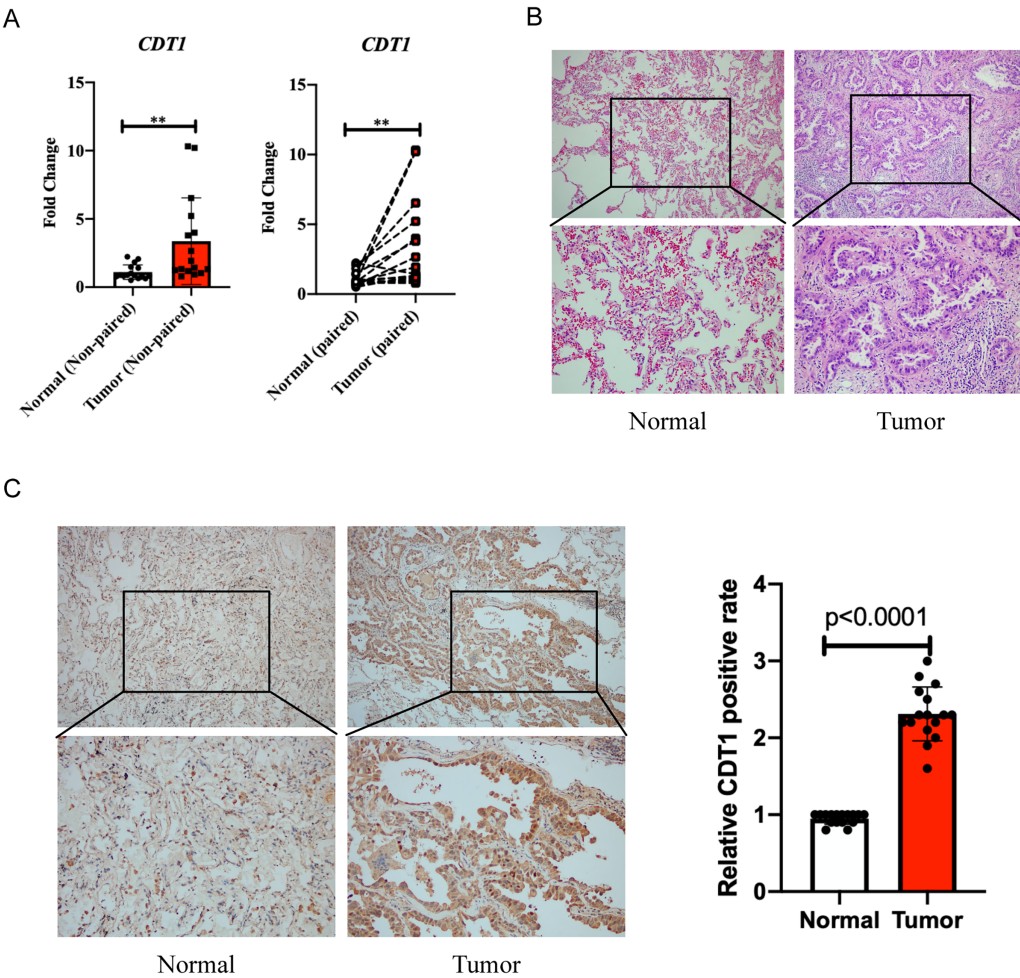

**Figure 3 Clinical samples to verify the expression level of CDT1 in LUAD and paired adjacent normal samples.** (A) qPCR showed an increased level of CDT1 in LUAD and paired adjacent normal samples (Normal group: $n$ = 16; Tumor group: $n$ = 16). (B) H&E staining for LUAD and paired adjacent normal samples (Normal group: $n$ = 16; Tumor group: $n$ = 16). (C) Immunohistochemistry detecting the expression of CDT1 in LUAD and paired adjacent normal samples (Normal group: $n$ = 16; Tumor group: $n$ = 16) (**$P$ < 0.01). CDT1, chromatin licensing and DNA replication factor 1; LUAD, lung adeno-carcinoma; qPCR, quantitative real-time PCR; H&E, hematoxylin, and eosin.

The LinkFinder module of the LinkedOmics database was used to explore the CDT1 co-expression network in LUAD. From this database, 3,182 genes were positively associated with CDT1. The Heatmap of the top 50 genes positively and negatively correlated with CDT1 are shown in Fig. 4C. The GO analysis indicated that CDT1 and the genes positively correlated with CDT1 were mainly involved in biological processes, including DNA replication, and cell cycle (Fig. 4D). The KEGG analysis indicated that CDT1 and the genes positively correlated with CDT1 were mainly involved in the cell cycle, DNA replication, and p53 signaling pathway (Fig. 4D).

It was obvious that the two analyses both aimed at DNA replication, and cell cycle, which were highly associated with LUAD progression.

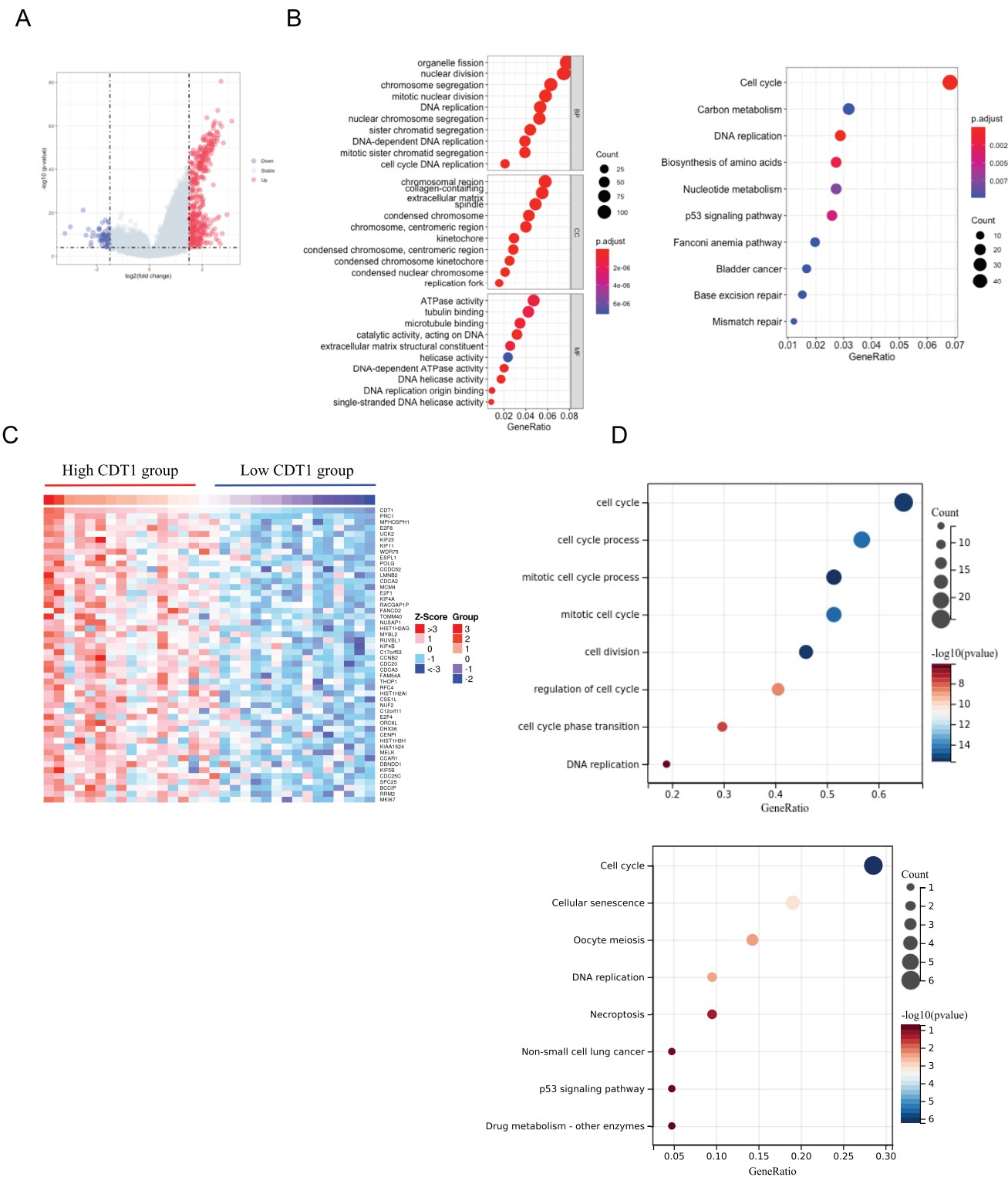

**Figure 4 CDT1 co-expressed genes and functional enrichment analysis.** (A) Volcano plot for DEGs between CDT1 high-expression and low-expression groups. (B) The GO and KEGG analysis for upregulated DEGs. (C) The heatmap for the top 50 genes that positively correlated with the expression of CDT1. (D) The GO and KEGG pathway analyses for CDT1 positive genes. CDT1, chromatin licensing and DNA replication factor 1; GO, gene ontology; KEGG, Kyoto Encyclopedia of Genes and Genomes; DEGs, Differentially Expressed Genes.

A

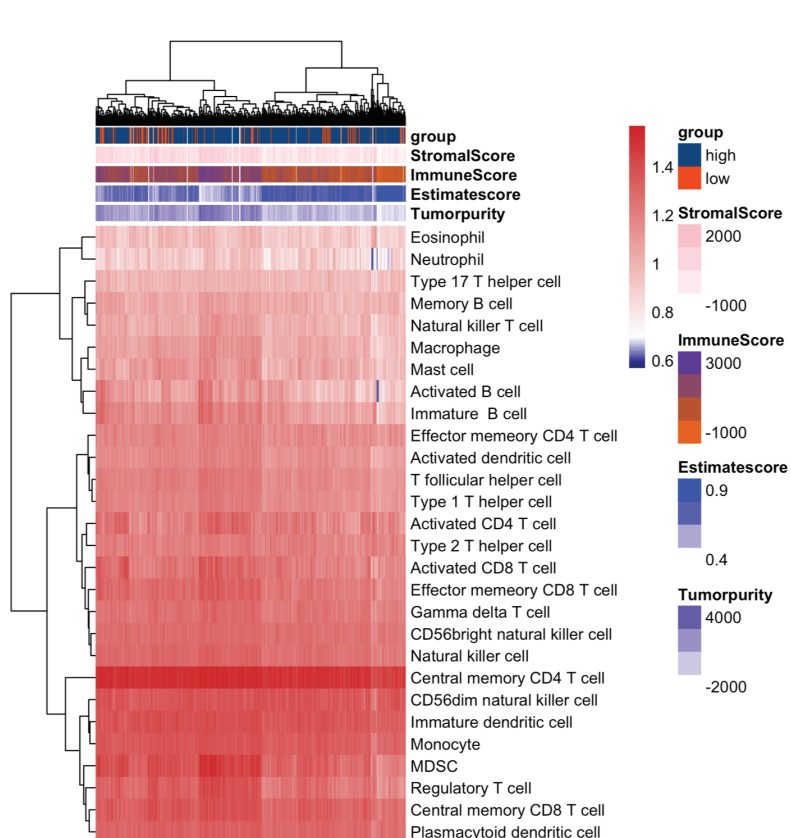

**Figure 5 (A) The landscape of immune cells using the ssGSEA scores based on ESTIMATEScore (CDT1 low group, *n* = 92; CDT1 high group, *n* = 417).** The correlation of CDT1 expression with the level of immune cell infiltration in LUAD. StromalScore: A computational algorithm that estimates the extent of stromal cell infiltration in tumor tissues. It quantifies the presence and abundance of stromal cells, which include fibroblasts, endothelial cells, and immune cells, within the tumor microenvironment. ImmuneScore: A computational metric that evaluates the level of immune cell infiltration in tumor tissues. It provides an estimation of the immune cell presence and activity within the tumor microenvironment. EstimateScore: A combined score that integrates both StromalScore and ImmuneScore. It represents the overall estimation of the stromal and immune components within the tumor, providing insights into the tumor microenvironment composition. Tumor purity: tumor purity refers to the proportion or percentage of cancer cells within the tumor sample. It represents the extent to which non-cancerous cells, such as stromal and immune cells, contribute to the overall tumor composition.

## Correlation analysis between CDT1 expression and immune infiltrates

In order to analyze the role of CDT1 expression in the tumor microenvironment (TME), the R 'GSVA' package and 'ESTIMATE' were used to analyze the relation among CDT1 expression, ESTIMATEScore, and immune infiltrates status. The landscape indicated that high expression of CDT1 was associated with higher tumor purity, lower ImmunScore, lower ESTIMATEScore as well as higher pro-tumor related immune cells infiltration (Fig. 5A).

In order to further confirm the relationship between CDT1 expression and immune cell infiltration, the R 'CIBERSORT' package was used to analyze the TME cell composition.

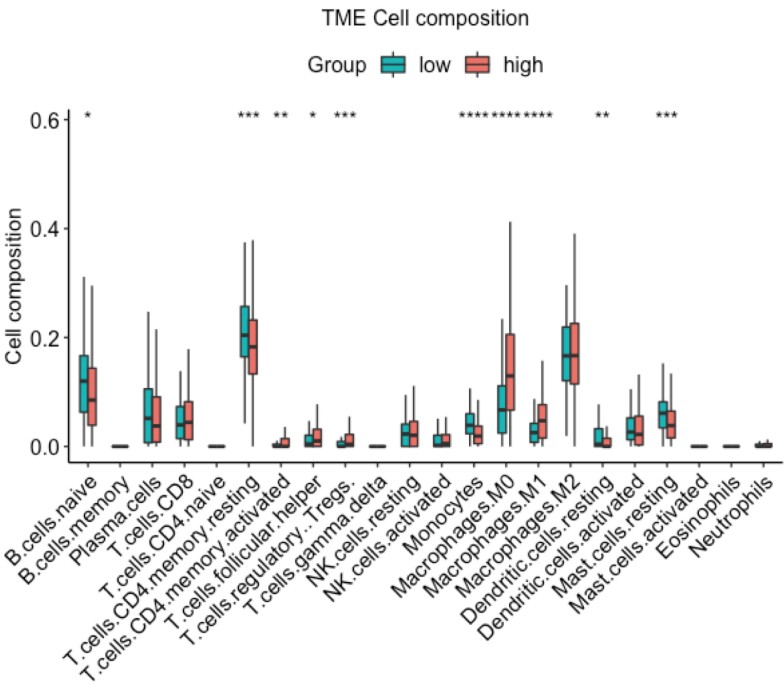

**Figure 6** **Box plot for the composition of 22 types of immune cells by CIBERSORT based on CDT1 low and high groups (Green: CDT1 low, *n* = 92; Red: CDT1 high, *n* = 417).** (\*$P < 0.05$, \*\*$P < 0.01$, \*\*\*$P < 0.001$, \*\*\*\*$P < 0.0001$).

The data (Fig. 6) demonstrated that high expression of CDT1 was significantly correlated with lower mast cells, and CD4 T cells expression. Earlier studies have shown that CD4 T cells promote pulmonary metastasis in mammary carcinomas by augmenting the protumor characteristics of macrophages (*DeNardo et al., 2009*). Mast cells can exert antitumor effects by stimulating immune responses, recruiting and activating other immune cells, and inducing tumor cell death (*Oldford & Marshall, 2015*). These findings suggest that the CDT1 high group is characterized by reduced mast cell expression and elevated CD4 T cell expression, indicative of a pro-tumor environment.

These data identified the positive relationship between a high expression level of CDT1 and a pro-tumor environment.

## CDT1-related immune genes predicted poor OS in LUAD patients

We further investigated the differentially expressed genes (DEGs) related to CDT1 in the immune response of lung adenocarcinoma (LUAD). The volcano plot (Fig. 7A) initially revealed the DEGs between the high and low ESTIMATEScore groups. Subsequently, a Venn plot (Fig. 7B) was employed to analyze the interaction between CDT1 and the immune response. From this analysis, five genes (Table 2) were selected for subsequent survival analysis. The survival analysis results (Fig. 7C) demonstrated that elevated expression levels of EPYC, EREC, GPR87, and OR2I1P were associated with poor OS in LUAD patients.

These findings indicate a correlation between the expression of CDT1-related immune genes and poor survival outcomes in patients with LUAD.

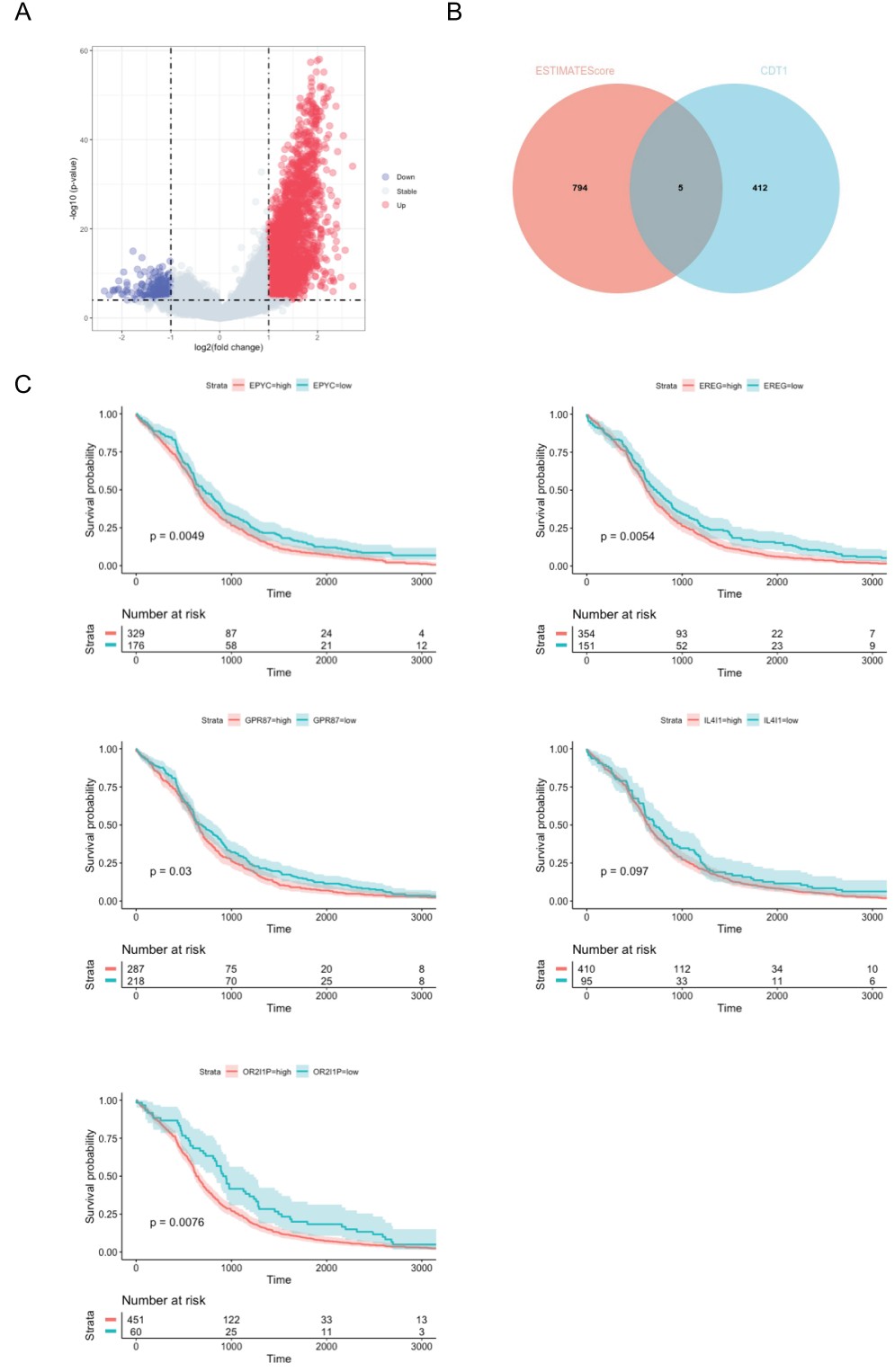

**Figure 7 CDT1-related immune genes predicted poot OS in LUAD patients.** (A) Volcano plot for DEGs between high ESTIMATEScore and low ESTIMATEScore groups. (B) Venn plot for the crosstalk. (C) Kaplan–Meier curves of OS of patients in the TCGA cohort (EPYC: high group, *n* = 329, low group, *n* = 176; EREG: high group, *n* = 354, low group, *n* = 151; GPR87: high group, *n* = 287, low group, *n* = 218; IL4I1: high group, *n* = 410, low group, *n* = 95; OR2I1P: high group, *n* = 451, low group, *n* = 60). Shadows:

**Figure 7** (continued)
The shadows represent the confidence intervals around the survival estimates. The shadows are typically shown as shaded areas around the plotted survival curves and provide information about the uncertainty associated with the survival estimates. The "Number at risk" tables accompanying the survival curves represent the number of individuals or cases still under observation at each time point.

**Table 2 CDT1-related immune genes.**

| No. | Gene | Gene full name | *P*-value |
|-----|------|----------------|-----------|
| 1 | IL4I1 | Interleukin-4-Induced-1 | 0.097 |
| 2 | OR2I1P | Olfactory receptor family 2 subfamily I member 1 pseudogene | 0.0076 |
| 3 | EPYC | Dermatan sulfate proteoglycan (Epiphycan) | 0.0049 |
| 4 | EREG | Epiregulin | 0.0054 |
| 5 | GPR87 | G protein-coupled receptor 87 | 0.03 |

## DISCUSSION

The most prevalent subtype of lung cancer, LUAD, is malignant and associated with a poor prognosis and high mortality rate. The need to identify novel prognostic indicators that can assist in the early diagnosis of LUAD and the development of therapeutic approaches for LUAD is driven by its debilitating characteristics.

Tumor genesis, invasion, and development are fundamentally influenced by aberrant DNA replication and cell cycle progression (*Zheng et al., 2020*). Numerous studies have shown that CDT1 is overexpressed in a variety of cancers, and this overexpression leads to low survival rates (*Mahadevappa et al., 2017*; *Seo et al., 2005*; *Li et al., 2021*; *Cao et al., 2021*). However, the exact mechanism underlying the action of CDT1 in LUAD remains unclear. We thoroughly examined CDT1 in LUAD in the current study, revealing CDT1 expression level, predictive and prognostic importance, function, and immune infiltration levels.

Consistent with previous reports, we found that (in the TCGA dataset) CDT1 expression levels were higher in LUAD and some other tumors than in normal tissues, where two GEO datasets were used to confirm our results. In addition, the high expression of CDT1 was highly correlated with poor prognosis, and patients in the TCGA dataset with high CDT1 expression had worse OS than those with low CDT1 expression. We further established a nomogram by integrating various parameters (including pathologic TNM stage, age, and CDT1 expression level) to predict individual patient mortality risk, which may help further therapy decisions. Finally, we obtained clinical samples from LUAD patients and performed qPCR and immunohistochemistry to investigate the expression level of CDT1 in LUAD and paired adjacent normal samples. Our data confirmed that CDT1 was increased in LUAD *vs* paired adjacent normal samples.

In addition, to explore the underlying mechanism of CDT1 and its co-expressed genes, a functional enrichment analysis was performed. GO enrichment results indicated that positively associated genes for CDT1 correlated with DNA replication, RNA localization, cell cycle DNA replication, and cell cycle checkpoints, while negatively associated genes for

CDT1 correlated with the regulation of T cell activation, leukocyte cell-cell adhesion, regulation of leukocyte proliferation, and T cell activation. KEGG analysis indicated that CDT1 and its co-expressed genes may play a vital role in the cell cycle—a finding that is in accordance with previous studies on CDT1 (*Kanellou et al., 2020*; *Fujita, 2006*). Moreover, the GO and KEGG enrichment for CDT1 expression level-related DEGs also indicated the vital role of the cell cycle in CDT1 high expression group, demonstrating the core effect of CDT1 in LUAD progression.

One of the most crucial elements of the tumor microenvironment during cancer development is the infiltration of host immune cells, which affects the growth and metastasis of lung cancer (*Van der Leun, Thommen & Schumacher, 2020*). We used 'ESTIMATE', 'CIBERSORT', and 'GSVA' to analyze the role of CDT1 in TME. Our data first indicated that CDT1 expression was positively associated with higher Tumorpurity, lower ImmuneScore as well as EstimateScore. In addition, the high expression of CDT1 was significantly correlated with lower mast cells, and CD4 T cells expression, which indicated high expression of CDT1 was associated with a pro-tumor environment. All these data indicate that high CDT1 expression is associated with infiltrating immune cells, which predicts poor survival of LUAD.

Finally, further investigation demonstrated the five core genes which play a function during CDT1-related immune reaction, and the survival analysis indicated the poor survival rate of these five genes in LUAD, demonstrating the poor role of CDT1-related immune genes in LUAD progression.

In our study, we focused on investigating the role of CDT1 in LUAD and its association with patient prognosis, immune infiltration, and functional enrichment analysis. While our findings provide valuable insights into the biological significance of CDT1 in LUAD, the direct clinical role and therapeutic implications of targeting CDT1 require further exploration.

Targeted drug design strategies aim to develop therapies that specifically target key molecules or pathways involved in tumor development and progression. Given the overexpression of CDT1 in LUAD and its association with poor prognosis, further investigation is warranted to evaluate CDT1 as a potential therapeutic target. Understanding the molecular mechanisms by which CDT1 promotes tumor growth and metastasis could provide valuable insights for developing targeted therapies.

Additionally, our functional enrichment analysis revealed the involvement of CDT1 and its co-expressed genes in key biological processes such as DNA replication and cell cycle regulation. These findings suggest that targeting CDT1-related pathways, such as the cell cycle, may hold promise for therapeutic intervention in LUAD. Further investigations into the downstream signaling pathways and molecular mechanisms influenced by CDT1 could uncover potential therapeutic targets within these pathways.

It is important to note that while our study provides a comprehensive analysis of CDT1 in LUAD, there are limitations that need to be addressed. We focused primarily on the expression level of CDT1 in LUAD samples and its correlation with clinical outcomes and immune infiltration. Future studies should consider conducting *in vitro* and *in vivo* experiments to validate the functional role of CDT1 in LUAD and explore potential

therapeutic interventions. Additionally, unraveling the underlying molecular mechanisms and downstream signaling pathways influenced by CDT1 would provide a more comprehensive understanding of its clinical relevance and potential therapeutic targets. What's more? We only verified the expression level of CDT1 in tumor tissues and normal tissues, the absence of experimental validations using MALDI-TOF mass spectrometry or immunoblotting limited our understanding of the deep molecular mechanism.

## CONCLUSIONS

In summary, our research found that CDT1 was upregulated in LUAD and that high CDT1 expression predicted poor prognosis. These results suggested that CDT1 may be a prognostic marker and therapeutic target for LUAD, which may help us to more precisely predict the survival and personalize the treatment for LUAD patients.

## ABBREVIATIONS

| | |
|---|---|
| **LUAD** | Lung Adenocarcinoma |
| **NSCLC** | Non-small Cell Lung Cancer |
| **CDT1** | Chromatin licensing and DNA replication factor 1 |
| **OS** | Overall Survival |
| **RT-qPCR** | Quantitative Real-Time Reverse Transcription Polymerase Chain Reaction |
| **IHC** | Immunohistochemistry |
| **H&E** | Hematoxylin and eosin |
| **ROC** | receiving operating characteristic |
| **DEGs** | Differentially Expressed Genes |
| **GEO** | Gene Expression Omnibus |
| **GSEA** | Gene Set Enrichment Analysis |
| **TCGA** | The Cancer Genome Atlas |
| **GO** | Gene Ontology |
| **KEGG** | Kyoto Encyclopedia of Genes |
| **TME** | Tumor Micro-environment |
| **TNM** | tumor-node-metastasis |
| **HZTCM** | Hangzhou Hospital of Traditional Chinese Medicine |
| **LAML** | Acute Myeloid Leukemia |
| **ACC** | Adrenocortical carcinoma |
| **BLCA** | Bladder Urothelial Carcinoma |
| **LGG** | Brain Lower Grade Glioma |
| **BRCA** | Breast invasive carcinoma |
| **CESC** | Cervical squamous cell carcinoma and endocervical adenocarcinoma |
| **CHOL** | Cholangiocarcinoma |
| **COAD** | Colon adenocarcinoma |
| **ESCA** | Esophageal carcinoma |
| **GBM** | Glioblastoma multiforme |
| **HNSC** | Head and Neck squamous cell carcinoma |

| | |
|---|---|
| **KICH** | Kidney Chromophobe |
| **KIRC** | Kidney renal clear cell carcinoma |
| **KIRP** | Kidney renal papillary cell carcinoma |
| **LIHC** | Liver hepatocellular carcinoma |
| **LUSC** | Lung squamous cell carcinoma |
| **DLBC** | Lymphoid Neoplasm Diffuse Large B-cell Lymphoma |
| **MESO** | Mesothelioma |
| **OV** | Ovarian serous cystadenocarcinoma |
| **PAAD** | Pancreatic adenocarcinoma |
| **PCPG** | Pheochromocytoma and Paraganglioma |
| **PRAD** | Prostate adenocarcinoma |
| **READ** | Rectum adenocarcinoma |
| **SARC** | Sarcoma |
| **SKCM** | Skin Cutaneous Melanoma |
| **STAD** | Stomach adenocarcinoma |
| **TGCT** | Testicular Germ Cell Tumors |
| **THYM** | Thymoma |
| **THCA** | Thyroid carcinoma |
| **UCS** | Uterine Carcinosarcoma |
| **UCEC** | Uterine Corpus Endometrial Carcinoma |
| **UVM** | Uveal Melanoma |

## ACKNOWLEDGEMENTS

We appreciate the efforts of our research group for helping in the completion of this article and the reviewers for reviewing this article. We would like to thank Editage for English language editing.

### Funding

This work was supported by the construction fund of Medical Key Disciplines of Hangzhou (2020SJZDXK004), the program of Zhejiang Provincial TCM Sci-tech Plan (2020ZQ041), and Zhejiang SL Famous Traditional Chinese Medicine Expert Inheritance Studio Project (GZS202002). The funders had no role in study design, data collection and analysis, decision to publish, or preparation of the manuscript.

### Grant Disclosures

The following grant information was disclosed by the authors:
Medical Key Disciplines of Hangzhou: 2020SJZDXK004.
Zhejiang Provincial TCM Sci-tech Plan: 2020ZQ041.
Zhejiang SL Famous Traditional Chinese Medicine Expert Inheritance Studio Project: GZS202002.

## Competing Interests

The authors declare that they have no competing interests.

## Author Contributions

- Jing Jiang conceived and designed the experiments, performed the experiments, prepared figures and/or tables, authored or reviewed drafts of the article, and approved the final draft.
- Yu Zhang conceived and designed the experiments, performed the experiments, prepared figures and/or tables, and approved the final draft.
- Jun Wang conceived and designed the experiments, performed the experiments, prepared figures and/or tables, and approved the final draft.
- Xuefei Yang performed the experiments, authored or reviewed drafts of the article, and approved the final draft.
- Xingchang Ren performed the experiments, authored or reviewed drafts of the article, and approved the final draft.
- Hai Huang performed the experiments, authored or reviewed drafts of the article, and approved the final draft.
- Jue Wang analyzed the data, prepared figures and/or tables, and approved the final draft.
- Jinhua Lu analyzed the data, prepared figures and/or tables, and approved the final draft.
- Yazhen Zhong analyzed the data, prepared figures and/or tables, and approved the final draft.
- Zechen Lin analyzed the data, prepared figures and/or tables, and approved the final draft.
- Xianlei Lin analyzed the data, prepared figures and/or tables, and approved the final draft.
- Yewei Jia conceived and designed the experiments, analyzed the data, prepared figures and/or tables, authored or reviewed drafts of the article, and approved the final draft.
- Shengyou Lin conceived and designed the experiments, authored or reviewed drafts of the article, funding support, and approved the final draft.

## Human Ethics

The following information was supplied relating to ethical approvals (*i.e.*, approving body and any reference numbers):

The research was performed in accordance with the Declaration of Helsinki.
The research methods in our study were approved by the Ethics Committee of HZTCM (No. 2021LH005). Informed consent was obtained from all patients.

## Data Availability

The datasets analyzed during the current study are available in the Cancer Genome Atlas: LUAD.

The data are also available at the Gene Expression Omnibus: GSE118370, GSE140797 and GSE31210.

## Supplemental Information

Supplemental information for this article can be found online at http://dx.doi.org/10.7717/peerj.16166#supplemental-information.

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
