# Peer review of "Identification of CDT1 as a prognostic marker in human lung adenocarcinoma using bioinformatics approaches"

_PeerJ, doi:10.7717/peerj.16166_

## Round 0.1 · original submission · Major Revisions

Please revise your manuscript according to the comments of our Reviewers.

Reviewer 1 ·

Basic reporting

no comment

Experimental design

no comment

Validity of the findings

no comment

Additional comments

85735v1
PeerJ
Identification of CDT1 as a prognostic marker in human lung adenocarcinoma using bioinformatics approaches

Thank you for the paper. This paper has found the prognostic significance of CDT1 for lung adenocarcinoma. Here are some suggestions for the authors, hoping to enrich the article.

1. All the current data presented and analyzed in this study were downloaded from publicly available databases, actually, only TCGA, GSE118370, and GSE140797. More data from multiple cohorts are needed to support the finding, for instance, other datasets from Gene Expression Omnibus (GEO), International Cancer Genome Consortium (ICGC), Cancer Cell Line Encyclopedia (CCLE), ONCOMINE, Genomic Expression Archive (GEA), ArrayExpress (EMBL_EBI), Chinese National Genomics Data Center (NGDC), and Sequence Read Archive (SRA).

2. The authors need to provide details for the dataset screening. How did the authors achieve GSE118370 and GSE140797? How about the inclusive and exclusive criteria? How about each step excluding other candidates? Please show the standard PRISMA flow. As far as I know, there are more microarray or RNA-sequencing datasets in GEO, ICGC, CCLE, GEA, ONCOMINE, ArrayExpress, NGDC, and SRA.

3. For an in-silico study, experimental validations should be available. The validation for the prognostic value with clinical samples from the authors’ institute is necessary, not only the expression level of CDT1 in lung adenocarcinoma tissues. The conclusion “CDT1 as a prognostic marker in human lung adenocarcinoma” needs to be supported by validation.

4. The standard mean deviation (SMD) and area under the curve (AUC) from the summary receiver operating characteristics (sROC) should be calculated to show the overall expression levels of CDT1 in lung adenocarcinoma by combining all possible datasets.

5. The summarized hazard ratios (HRs) should be calculated to show the prognostic roles combining all possible datasets all over the world, too.

6. Matrix-assisted laser desorption/ionization (MALDI) - time-of-flight (TOF) mass spectrometry, or immunoblotting could be considered for protein detection.

7. The functional experiments are also encouraged.

8. In the discussion section, the authors should discuss both the internal and external threats to the validity of this study, as well as the molecular mechanisms deeper.

9. The following papers could be compared and discussed.

Intermittent hypoxia-induced downregulation of microRNA-320b promotes lung cancer tumorigenesis by increasing CDT1 via USP37.
Mol Ther Nucleic Acids 10.183 2021
Li W, Huang K, Wen F, Cui G, Guo H, He Z, Zhao S.
DOI: 10.1016/j.omtn.2020.12.023
PMID: 33898105

SIRT3 promotion reduces resistance to cisplatin in lung cancer by modulating the FOXO3/CDT1 axis.
Cancer Med 4.711 2021
Cao Y, Li P, Wang H, Li L, Li Q.
DOI: 10.1002/cam4.3728
PMID: 33655712

Reviewer 2 ·

Basic reporting

Jiang et al presented a bioinformatics based study to showcase the role of CDT1 in early detection of lung adenocarcinoma. The study is based on publicly available cancer-specific datasets such as TCGA and GEO. The main claims of the paper is derived from the expression levels of CDT1 and related prognostic and diagnostic scores.

I have several concerns about the current manuscript,

1. My biggest concern with the paper is reproducibility. As the main novelty of the paper is from the statistical analysis of the datasets. The scripts that are used to reproduce the paper are of utmost importance. However I don’t see any link to GitHub or any other files containing the code to reproduce the claims. The softwares that are mentioned in the manuscript are open-source, however their parametric settings change the outcome. I suggest the authors to compile all the source code in R and provide an end-to-end notebook that can reproduce all the plots in the paper.

2. It seems the authors used both publicly available datasets and also produced human tissue samples from Lung adenocarcinoma. But it’s not clear where the new dataset is deposited. Will there be a GEO link available with the data?

3. It is a well-known fact that the expression of CDT is in general up-regulated in tumor datasets. Largely the Figure 1 is a reproduction of that known claim. I feel it’s a bit redundant to depict that finding. I would suggest authors to use non-LUAD cancers to be used as a second control to show indeed the expression of CDT1 is specific to LUAD and is significantly high.

4. Figure 1A shows highest expression of CDT1 in LAML. I am curious how do that match with the rest of the narrative of the manuscript. The authors should investigate and describe if there is anything important about that high expression compared to LUAD samples.

5. Figure 2A depicts the survival plots, otherwise known as Kaplan-Meier plot. While discussing the plot the authors wrote, “The survival rate of patients with LUAD was calculated using R program software ” — this description is insufficient and very generic. I suggest the authors to describe the procedure and provide source code for the same.

6. Figure 4 shows the associated pathways of the tumor samples that shows high expression of CDT1. The figures are however poorly annotated. For example, Figure 4 C does not have any annotation for x axis. Figure 4D bar-plots does has their x-axis squashed and therefore not visible.

7. The interpretation and motivation of Figure 6A is not clear to me. In line 247-249 the authors mention the high expression of CD4T cell expression and it indices high expression of CDT1 is associated to pro-tumor environment. I am not sure how this conclusion is derived.

8. The clinical role of CDT1 is not clear from the analysis. Can the authors also describe how this finding can be helpful for targeted drug-design and related pathways that can be of interest.

Overall the paper lacks professional description of the method section. The statistical analysis is for Figure 2,3 are not clearly written and described without any details. The comprehension can be improved by improving the writing. For example the sentence in line 48-49 is grammatically wrong. The figures are not always readable. The resolution of the axis ticks are not legible, I suggest authors to regenerate the plots with better resolution by using R function and raster graphics.

The discussion section is well-written and summarized the paper. The main weakness of the paper is lack of details in the statistical analysis and elaborate method description. Corroborating the findings with cell-type specific analysis with existing single-cell RNA-seq (scRNA-seq) analysis might improve the acceptability of the paper.

Experimental design

no comment

Validity of the findings

Without reproducible code to regenerate the figures it's hard to test the validity of the manuscript. A github repository with the end-to-end code for regenerating the plot can help to validate the claims from the manuscript.

Additional comments

no comment

Reviewer 3 ·

Basic reporting

See the Additional Comments.

Experimental design

See the Additional Comments.

Validity of the findings

See the Additional Comments.

Additional comments

The authors combined bioinformatics, qPCR, and IHC to explore the role of CDT1 in the pathogenesis and prognosis of lung adenocarcinoma. Overall, this study is suitable for publication, only if the authors address the following issues:

1. Throughout the manuscript, it seems better to use Grammarly (https://www.grammarly.com/) to check & correct potential grammatical errors or typos. For example,
1.1 In the Results of ABSTRACT, it seems better to change "... , the level of CDT1 expression was much higher in LUAD tissue than in healthy lung tissue" into "... . As to lung cancer, the level of CDT1 expression was much higher in LUAD tissue than in healthy lung tissue", which would be clearer and more cohesive (that is, sentences are closely connected).
1.2 In RESULTS' "3.1. CDT1 expression was upregulated in LUAD, based on the TCGA and GEO databases", it would be more accurate to rewrite "In order to show the expression of CDT1 in LUAD more clearly."

2. In all FIGURES, it would be clear and more readable to expand on figure legends by explaining the meanings of colors, groups, lines, and abbreviations. For example,
2.1 In the legends of all figures, it would be more rigorous to mention the sample size.
2.2 In the legend of Figure 1A, it would be more informative to list the full names of the cancer abbreviations.
2.3 In Figure 1B, it would be clearer and more informative to label which "normal" was non-paired or paired.
2.4 In Figure 1D, it would be clearer to highlight the dot of CDT1 using a different color.
2.5 In the legend of Figure 2, it would be more informative to explain the meanings of all elements (different colors, columns, shadows, the tables of "Number at risk", etc.) AND cite a reference (PMID: 28624402), which showed how to interpret or utilize a nomogram.
2.6 In the legend of Figure 3, it would be more accurate to mention "paired and non-paired". Likewise, please change the x-axis and legend of Figure 3A.
2.7 In the legend of Figure 5, it would be more informative to explain the meaning of "StromalScore", "ImmuneScore", "EstimateScore", and "Tumorpurity". Ideally, please mention these in METHODS.
2.8 In Figure 7, it would be more informative to explain the meanings of both shadows and "Number at risk" tables.

These revisions would greatly help readers, who do not specialize in bioinformatics, to understand the results and their implications easily and efficiently.

3. In ABSTRACT:
3.1 In Background, it would be more informative to rewrite "Chromatin licensing and DNA replication factor 1 (CDT1) is essential for both cell cycle control and replication in eukaryotic cells" by mentioning the relationship between CDT1 and cancer. This revision would help justify why the authors focused on CDT1 rather than any other gene.
3.2 In Background, it would be more accurate to rewrite "This study used bioinformatics techniques to assess CDT1 expression and its predictive value in lung adenocarcinoma (LUAD)" by mentioning the technique apart from "bioinformatics techniques", because the authors also used molecular experiments.
3.3 In Methods, it would be more informative and rigorous to expand on "For bioinformatics analysis, TCGA, GEO, LinkedOmics, GO, KEGG, TIMER, Nomogram, and Kaplan-Meier plotters were used" by mentioning the specific purpose of using these tools. For example, GO and KEGG are usually used to predict the biological processes as well as signaling pathways, in which some genes could participate.
3.4 In Methods, it seems better to change "To assess the amounts of RNA and protein expression in LUAD and paired adjacent normal tissues, qPCR and immunohistochemistry were used" into "To assess the RNA as well as protein expression of CDT1 in both LUAD and paired adjacent normal tissues, qPCR and immunohistochemistry were used", which would be more informative and rigorous.
3.5 In Results, it seems better to change "CDT1 was upregulated in pan-cancers based on the TCGA and GEO datasets" into "CDT1 was upregulated in the vast majority of cancer tissues, based on pan-cancer analysis in TCGA and GEO datasets", which would be more rigorous.
3.6 In Results, it would be more informative to expand on "Our clinical data supported these findings" by mentioning what was the "clinical data".
3.7 In Results, it would be more informative and clearer to expand on "The Kaplan3Meier survival curve revealed poor survival rates in CDT1 high expression group than the low expression group" by mentioning how the authors obtained "CDT1 high expression group" vs "the low expression group".
3.8 In Results, it seems better to change "To determine if CDT1 expression was an independent factor in LUAD patients" into "To determine if CDT1 expression was an independent risk factor in LUAD patients", which would be more accurate.
3.9 In Results, it seems better to change "Univariate and multivariate Cox regression analysis showed that CDT1 was a potential novel prognosis factor for LUAD patients" into "Univariate and multivariate Cox regression analysis showed that CDT1 was a potential novel prognosis factor for LUAD patients, whose prognosis was poorer when CDT1 expression was higher", which would be easier to understand.
3.10 In Results, it seems better to change "Functional enrichment analysis indicated that CDT1 was involved in the cell cycle" into "Based on functional enrichment analysis, highly expressed DEGs of CDT1-high patients were predicted to be involved in cell cycle", which would be more informative and rigorous. Likewise, please rewrite "According to an investigation of immune infiltration, CDT1 substantially correlated with immune cell subsets and predicted a low chance of survival in LUAD patients".
3.11 In Results, it would be more informative and accurate to expand on "According to an investigation of immune infiltration, CDT1 substantially correlated with immune cell subsets and predicted a low chance of survival in LUAD patients" by mentioning the specific types of "immune cell" substantially correlated with high CDT1 expression.\
3.12 In Conclusions, it would be more concise to delete "Our data indicated comprehensively and systematically analyzed the expression level in the datasets as well as in our own clinical samples, we also evaluated the prognostic and diagnostic value of CDT1, and finally, the potential mechanisms of CDT1 in the progression of LUAD".
3.13 In Conclusions, it seems better to change "All these results suggested that CDT1 may be a prognostic marker and therapeutic target for LUAD" into "These results suggested that CDT1 may be a prognostic marker and therapeutic target for LUAD", which would be clearer and more cohesive.

4. In INTRODUCTION:
4.1 In Paragraph 2, it seems better to change "The precise roles and prognostic relevance of CDT1 in the evolution of LUAD are still unknown; nonetheless, knowing CDT1 function is essential if we are to enhance our understanding of cancer transformation and progression" into "However, the precise roles and prognostic relevance of CDT1 in the evolution of LUAD are still unknown. Knowing CDT1 function is essential for us to better understand cancer transformation and progression", which would be clearer and more cohesive.
4.2 In Paragraph 3, it seems better to change "we comprehensively explored CDT1 expression, function, prognostic value, and immune infiltration in LUAD through bioinformatics analyses" into "we comprehensively explored CDT1's expression patterns, potential function, prognostic value, and relationship with immune infiltration in LUAD through bioinformatics analyses", which would be more accurate.

5. In RESULTS, it would be clearer to end each paragraph in RESULTS with one sentence: "Together, these results suggest that ..." (a pattern like PMID: 34715879, PMID: 34384362, PMID: 35965679, and PMID: 34537192), summarizing a paragraph AND highlighting the implications of all results in the paragraph.

6. In CONCLUSIONS, it would be more concise to delete "Our data comprehensively and systematically analyzed CDT1 expression levels in separate datasets and in our own clinical samples. We also evaluated the prognostic and diagnostic value of CDT1. Finally, the potential mechanisms of CDT1 in the progression of LUAD were also investigated".

7. In SUPPLEMENTAL MATERIAL, it seems better to add an English version of "peerj-85735-090-the_original_ethics_approval_form", because this revision would help international readers understand.

---

## Round 0.2 · Major Revisions

Thank you for responding and making revisions, but the current status of the paper is still not satisfactory to the 2 Reviewers. Please revise and respond based on the Reviewers' valuable comments, especially the first Reviewer's suggestions.

Reviewer 2 ·

Basic reporting

The authors have addressed the concerns raised in the previous manuscript. However there are some points which still remain,
1. The authors mention that their own experiment is not sharable and is not RNA-seq in nature. In light of that they should increase the number of datasets and cohorts to increase the scope of study. In the interest of time and feasibility I suggest adding at least another publicly available single-cell dataset in the comparison. The dataset the authors have chosen do not include any clinical information therefore the validity of being prognostic marker is not conclusively validated. That being the main claim of the paper should be established.

Experimental design

no comment

Validity of the findings

2. The authors mention in their rebuttal,
"While we acknowledge the potential value of including non-LUAD cancers as an additional control group, it is important to note that our study was specifically focused on LUAD. Incorporating other cancer types could introduce confounding factors and dilute the specificity of our findings."

I don't see how this statement is valid. If CDT1 is a prognostic marker for LUAD, then it would show significance is detecting LUAD in samples. For example, if CDT1 is also a prognostic marker for a few other types of cancers, then it's hard to conclusively argue that it's a marker for only LUAD.

Additional comments

The reviewers have provided the github repo. The scripts are deposited, but could not be run as they use local directory paths which are not absolute. I would recommend adding commands like
`wget` with a README.md file in gitgub repo that would describe the download of the datasets.

Reviewer 3 ·

Basic reporting

Thank the authors for their efforts to respond to all of my comments. Overall, this version would be suitable for publication, only if the authors address the following issues, to which the authors did not seem to adequately respond :

1. My previous comment 1.2 (In RESULTS' "3.1. CDT1 expression was upregulated in LUAD, based on the TCGA and GEO databases", it would be more accurate to rewrite "In order to show the expression of CDT1 in LUAD more clearly."). To be cleaerer, the sentence seems grammatically incorrect. It would be more accurate to change "In order to show the expression of CDT1 in LUAD more clearly. The TCGA database's paired tumor and normal nearby samples, as well as the unpaired samples, were also studied using the two different statistical analysis techniques" into "In order to show the expression of CDT1 in LUAD more clearly, the TCGA database's ...".

2. My previous comment 2.1 (2.1 In the legends of all figures, it would be more rigorous to mention the sample size). The authors mentioned the sample size in only Figure 3. It would be more informative to state the sample size in ALL FIGURES' legends.

3. My previous comment 2.5 (2.5 In the legend of Figure 2, it would be more informative to explain the meanings of all elements (different colors, columns, shadows, the tables of "Number at risk", etc.) AND cite a reference (PMID: 28624402), which showed how to interpret or utilize a nomogram.). The authors did not seem to provide new, informative changes to the legend of Figure 2. The changes to "2.4. Identification of Differentially Expressed Genes and Building a nomogram" did not seem to successfully help readers to understand how to interpret a nonogram. Please refer to PMID: 2862440 to find out useful expressions to describe elements in a nonogram.

Experimental design

N/A

Validity of the findings

N/A

Additional comments

N/A

---

## Round 0.3 · accepted · Accept

Despite not being able to get a response from 2 reviewers, another reviewer has approved the revised paper for publication. I was satisfied with the responses and revisions made by the authors. The Reviewer's and my concerns have been well addressed. I believe that this revised manuscript is ready to be considered for publication in this journal.

Reviewer 3 ·

Basic reporting

Thank the authors for their efforts to respond to all of my comments. The current manuscript would be suitable for acceptance.

Experimental design

N/A

Validity of the findings

N/A

Additional comments

N/A